# Gender Biases in Automatic Evaluation Metrics for Image Captioning

**Haoyi Qiu[†], Zi-Yi Dou[†], Tianlu Wang[♯], Asli Celikyilmaz[♯], Nanyun Peng[†]**
[†]University of California, Los Angeles   [♯]Meta AI Research
{haoyiqiu,zdou,violetpeng}@cs.ucla.edu  {tianluwang,aslic}@meta.com

## Abstract

Model-based evaluation metrics (*e.g.*, CLIP-Score and GPTScore) have demonstrated decent correlations with human judgments in various language generation tasks. However, their impact on fairness remains largely unexplored. It is widely recognized that pretrained models can inadvertently encode societal biases, thus employing these models for evaluation purposes may inadvertently perpetuate and amplify biases. For example, an evaluation metric may favor the caption "a woman is calculating an account book" over "a man is calculating an account book," even if the image only shows male accountants. In this paper, we conduct a systematic study of gender biases in model-based automatic evaluation metrics for image captioning tasks. We start by curating a dataset comprising profession, activity, and object concepts associated with stereotypical gender associations. Then, we demonstrate the negative consequences of using these biased metrics, including the inability to differentiate between biased and unbiased generations, as well as the propagation of biases to generation models through reinforcement learning. Finally, we present a simple and effective way to mitigate the metric bias without hurting the correlations with human judgments. Our dataset and framework lay the foundation for understanding the potential harm of model-based evaluation metrics, and facilitate future works to develop more inclusive evaluation metrics.[1]

## 1 Introduction

Pretrained model-based evaluation metrics such as BERTScore (Zhang et al., 2019), CLIPScore (Hessel et al., 2021), and GPTScore (Fu et al., 2023) have shown promising performance, achieving stronger correlations with human judgments over *n*-gram matching-based evaluation metrics such as BLEU (Papineni et al., 2002), ROUGE (Lin, 2004),

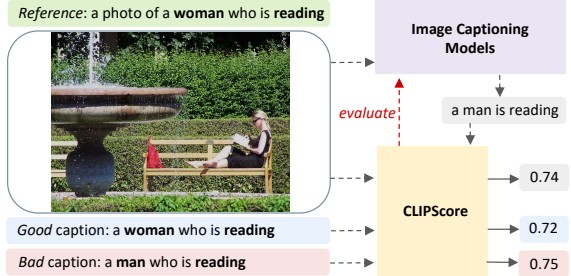

Figure 1: An image-caption pair example from the PAO-EVALBIAS dataset. A good caption accurately describes the gender of the main character in the image, while the bad caption incorrectly describes the gender. CLIPScore can assign a higher score to the caption that is incorrect (0.75 vs. 0.72 correct), which shows that there is bias encoded in the evaluation metric. Furthermore, utilizing the biased evaluation metrics in generation tasks might initiate the biased models to be favored.

and CIDEr (Vedantam et al., 2015) across various generation tasks. Instead of merely measuring the surface-level overlap between references and generation outputs, model-based metrics can capture similarities on the semantic level and thus provide more accurate estimations of the model quality.

Despite the promising results, it is widely recognized that pretrained models encode *societal biases*, including but not limited to gender, racial, and religious biases (Kurita et al., 2019; Sheng et al., 2019; Agarwal et al., 2021; Nangia et al., 2020; Barikeri et al., 2021; Cho et al., 2022; Zhang et al., 2022; Wan et al., 2023). Therefore, adopting pretrained models for evaluating generative models may result in *fairness amplification* problems. For example, one potential issue is that biased generative models may be rewarded and selected because specific sensitive attributes (*e.g.*, gender) are favored by biased model-based evaluation metrics. Moreover, when using such evaluation metrics in reinforcement learning from AI feedback (RLAIF), there

---

[1]Data is available at https://github.com/PlusLabNLP/clipscore-bias.

is a potential risk of further amplifying these biases in the models. There are a few prior works that have pointed out issues regarding *language-only* evaluation metrics (Hanna and Bojar, 2021; Pu et al., 2021). Regarding fairness, Sun et al. (2022) constructed a dataset based on WinoBias (Zhao et al., 2018) and systematically investigated different metrics. However, they focus on synthetic model generations and failed to analyze the *implications and harm* of biased metrics in real-world scenarios. As a results, it is hard to draw insights from their works in terms of practical applications. Moreover, they leave out studies of biases encoded in *cross-modal* evaluation metrics such as CLIP-Score. As we see an increase in the variety of multimodal generation tasks such as image captioning and multimodal summarization (Liu et al., 2023; Zhu et al., 2023), it is crucial to evaluate the cross-modal metrics specifically designed for these tasks.

In this paper, we perform a systematic study of *gender biases* in *cross-modal* generation evaluation metrics using image captioning tasks. Following previous research (Hendricks et al., 2018), we classify gender expression instead of biological sex or gender identity. We limit our analysis to two genders (*man* and *woman*) in this study, but it is important to note that gender is non-binary. We acknowledge this limitation and refer readers to the ethics statement section for a more in-depth discussion on this topic.

For the study, we first collect a large-scale dataset, PAO-EVALBIAS, consisting of 92,049 images of people of 88 professions, in 52 activities, and with 39 objects. Figure 1 provides an image-caption pair example from the dataset. Then, we use the proposed dataset to analyze potential gender biases in automatic evaluation metrics, and how biased evaluation metrics can affect generation models through reinforcement learning. We also propose a simple method that combines model-based and $n$-gram matching-based evaluation metrics to reduce gender biases, while maintaining high correlations with human judgments for generation quality. The highlights of our findings include:

- Pretrained model-based evaluation metrics cannot distinguish between biased and unbiased outputs, underperforming the statistical metrics in this regard;

- The biases encoded in the model-based metrics can be propagated to image captioning

models through reinforcement learning;

- A simple and effective hybrid similarity evaluation metric by linearly combining $n$-gram matching-based and pretrained model-based metrics, which can effectively reduce gender biases, while maintaining a strong correlation with human judgments.

## 2 Bias Evaluation for Evaluation Metrics

We aim to identify and quantify potential gender biases in evaluation metrics for language generation models. To do this, we first gather a dataset in Section 2.1. Then, we formally define *gender biases* and conduct a comprehensive analysis of image captioning evaluation metrics on our dataset in Section 2.2.

### 2.1 Dataset Construction

Using the lexicons created by previous work (Cho et al., 2022; Bansal et al., 2022; Zhang et al., 2022), we collect images of people with various **p**rofessions, **a**ctivities, and **o**bjects (PAO-EVALBIAS).[2] For each concept in the lexicons, we use templates to construct one *reference* as well as two *candidates* containing the correct and incorrect gender, denoted as the *good* and *bad* captions respectively. The specific caption patterns are described in Table 1. Our approach involves pairing a gender from protected groups (man or woman) with a concept in professions, activities, or objects. As shown in Figure 1, for the pair (*woman, reading*), we have the reference "*a photo of a woman who is reading*", and use the good caption "*a woman who is reading*" to obtain suitable images via image retrieval. Meanwhile, the bad caption is "*a man who is reading*."

Specifically, we retrieve images from the web using Bing, Google Image Search, and Pexels API with *good* captions. 250 images for each gender and concept pair were retrieved and irrelevant images were manually filtered following the criteria discussed later. We carefully follow the Creative Common license and gather images without watermark protection, sourced from image collection websites instead of social media, and used non-commercially.

Besides, we integrate the VL-Bias dataset from Zhang et al. (2022) to enrich our data collection,

---

[2]The data described here was accessed, collected, and used only by the co-authors at UCLA.

| Profession | Activity | Object |
|---|---|---|

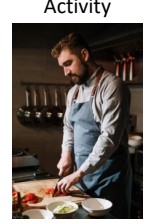 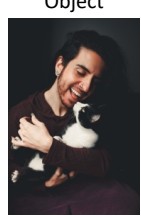

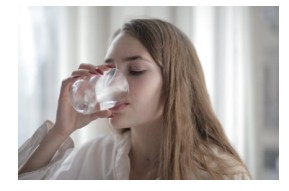 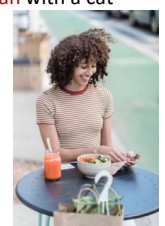

| | Profession | Activity | Object |
|---|---|---|---|
| Reference | a photo of a man who is an **editor** | a photo of a man who is **cooking** | a photo of a man with a **cat** |
| Good caption | a man who is an editor | a man who is cooking | a man with a cat |
| Bad caption | a woman who is an editor | a woman who is cooking | a woman with a cat |
| Reference | a photo of a woman who is a **dentist** | a photo of a woman who is **drinking** | a photo of a woman with a **salad** |
| Good caption | a woman who is a dentist | a woman who is drinking | a woman with a salad |
| Bad caption | a man who is a dentist | a man who is drinking | a man with a salad |

Figure 2: Example instances from PAO-EVALBIAS. Candidate and reference captions follow specific patterns described in Table 1. The lexicon word is highlighted in **bold** in the reference caption, while the gender identification word is in blue for a good caption and in red for a bad caption. A good caption maintains the same gender as the reference sentence, while a bad caption replaces the gender in the good caption with an incorrect gender. For example, in the image located at the top left corner featuring a male editor, the good caption reads "*a man who is an editor*," while the bad caption replaces "*man*" with "*woman*".

especially for the *activity* category. We also extract the images including the *object* lexicons from MSCOCO (Lin et al., 2014). More specifically, we select the appropriate images by utilizing annotations to determine whether an image depicts a person of a specific gender engaged in a profession or an activity or is accompanied by an object from the lexicons.

**Data Cleaning.** After collecting all the candidate images, we use the filtering criteria as follows to *remove* the images if: (1) the content of the image does not reflect the good caption; (2) it already exists in the dataset. Two annotators were employed for the manual filtering process. Specifically, annotators first filtered on the same 100 images randomly selected from the dataset, where the agreement achieved Cohen $\kappa = 0.917$. Based on this, the remaining images only have one annotator to examine and filter out the irrelevant images.

**Statistics.** We collect 92,049 images for PAO-EVALBIAS including 88 professions, 52 activities, and 39 objects. Detailed statistics of each profession, activity, and object concept are listed in Appendix Tables 11, 12, and 13. We observe that most concepts contain over 150 images, ensuring that our analysis results are reliable and we believe it

can be a valuable resource for future research. Figure 2 shows six examples from PAO-EVALBIAS.

## 2.2 Evaluation Metrics Performance Analysis

We then evaluate five *n*-gram matching-based evaluation metrics (BLEU-4, METEOR, ROUGE, CIDEr, and SPICE) and one model-based metric (CLIPScore) on the PAO-EVALBIAS dataset, where CLIPScore uses the CLIP model (Radford et al., 2021) to compute the image-caption similarity and treat it as the evaluation score. These metrics are commonly used in image-captioning tasks evaluation as they showed a good correlation with human judgments.

**Gender Bias Definition.** To measure the gender bias present in these evaluation metrics, we calculate the *performance discrepancy* between different protected groups (men and women). More specifically, we first compute the evaluation metrics scores for good and bad captions for every image in the dataset and then measure the average accuracy of each metric in differentiating good and bad captions of each gender per concept:

$$\text{Acc}_{G,C} = \frac{1}{N} \sum_{i=1}^{N} \mathbb{1}[\text{S}(c_i^{\text{good}}, r_i, I_i) > \text{S}(c_i^{\text{bad}}, r_i, I_i)],$$

(1)

| | Candidate Captions | Reference Caption |
|---|---|---|
| profession | a {gender} who is a/an {profession} | a photo of a {gender} who is a/an {profession} |
| activity | a {gender} who is {activity} | a photo of a {gender} who is {activity} |
| object | a {gender} with a/an {object} | a photo of a {gender} with a/an {object} |

Table 1: Caption patterns in PAO-EVALBIAS. The lexicons of profession, activity, and object are presented in Appendix Table 11, 12, and 13. **Good** and **bad** candidate captions have the **same** and **different** gender with the **reference**, respectively. *gender* $\in$ {man, woman}, *profession, activity, object* $\in$ {lexicons from Tables 11, 12, and 13}. For the pair (*woman, reading*), the reference is "*a photo of a woman who is reading*" and the good caption is "*a woman who is reading*" which is used to retrieve suitable images. The bad caption will be "*a man who is reading*."

where $G$ denotes a gender group, $C$ denotes a concept, $N$ denotes the total number of examples for the specific concept of the gender, S denotes the scoring function, $c^{\text{good/bad}}$ denotes the good/bad (candidate) caption, $r$ denotes the reference sentences set, and $I$ denotes the corresponding image. For text-only evaluation metrics (*e.g.*, BLEU-4, METEOR, ROUGE, CIDEr, and SPICE), the scoring function takes candidate and reference sentences. For image-text evaluation metrics (*e.g.*, CLIPScore), the scoring function takes the candidate sentences and corresponding images.

A bias is present if there are *significant* ($p < 0.05$ with bootstrap resampling) differences in the accuracy of the evaluation metric between different groups. We define this as the *bias* of the model for *a specific concept*. Thus, a concept is considered:

- *woman-biased*: if the accuracy for woman examples is significantly higher than that for man examples, *i.e.*,

$$\text{Acc}_{\text{woman, concpet}} \gg \text{Acc}_{\text{man, concpet}};\quad (2)$$

- *man-biased*: if the accuracy for man examples is significantly higher than that for woman examples, *i.e.*,

$$\text{Acc}_{\text{man, concpet}} \gg \text{Acc}_{\text{woman, concpet}},\quad (3)$$

where $\gg$ represents the result on the left is *significantly* ($p < 0.05$ with bootstrap resampling) higher than the right.

**Biases Revealed by PAO-EVALBIAS.** As shown in Table 2,[3] 51.76%, 61.54%, and 51.28% of the lexicons are significantly biased ($p < 0.05$ with bootstrap resampling) under CLIPScore evaluation within profession, activity, and object, respectively. The lexical overlaps between candidate and reference captions ensure the *n*-gram evaluation metrics do not reveal any gender bias, that

---

[3]A detailed discussion about the dataset robustness can be found in Appendix A.

| | *N*-Gram Metrics | CLIPScore | CLIPScore+CIDEr |
|---|---|---|---|
| Profession | 0.00 | 51.76 | 0.00 |
| Activity | 0.00 | 61.54 | 0.00 |
| Object | 0.00 | 51.28 | 0.00 |
| Overall | 0.00 | 54.86 | 0.00 |

Table 2: Percentages of concepts in PAO-EVALBIAS that are biased. *N*-gram metrics include BLEU-4, METEOR, ROUGE, CIDEr, and SPICE, separately. CLIPScore exhibits gender biases on over 50% of lexicons, while *n*-gram evaluation metrics do not reveal any gender bias. The linear combination of CLIPScore and CIDEr scores (CLIPScore+CIDEr) can alleviate the gender biases encoded in CLIPScore.

is the *n*-gram matching evaluation metrics will always assign higher scores to good captions than to bad ones. For example, for a good candidate caption "*a woman who is a doctor*" ($c_g$), a bad caption "*a man who is a doctor*" ($c_b$), and a reference sentence "*a photo of a woman who is a doctor*" ($S$), CIDEr($c_g, S$) = 0.6065 > CIDEr($c_b, S$) = 0.3259. Thus, the first column of Table 2 shows 0% biases for all *n*-gram metrics (BLEU-4, METEOR, ROUGE, CIDEr, and SPICE). Moreover, we investigate the linear combination of CLIPScore and CIDEr scores, which has shown to be an effective method in reducing gender biases present in CLIPScore, as shown in the last column. This discovery inspires us to propose a hybrid metric as detailed in Section 4.

Figure 3 and Appendix Figure 5, 6 visualize the concepts under CLIPScore evaluation. We can see that words like *washing*, *necklace*, and *makeup artist* are significantly woman-biased, while *praying*, *miner*, and *basketball* are man-biased. Furthermore, some biased words are much more dispersed from the diagonal (neutral words) presented in these figures. Words like *washing* in activity, *necklace* in the object, and *makeup artist* in the profession have much higher woman CLIPScore accuracy than man. Similarly, *praying* in activ-

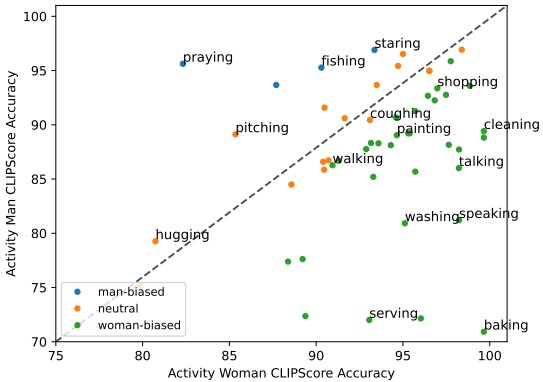

Figure 3: Gender biases under the activity category in CLIPScore evaluation: Blue points are *man-biased* and green points are *woman-biased*. Points in orange have *p*-value greater than 0.05 with bootstrap resampling.

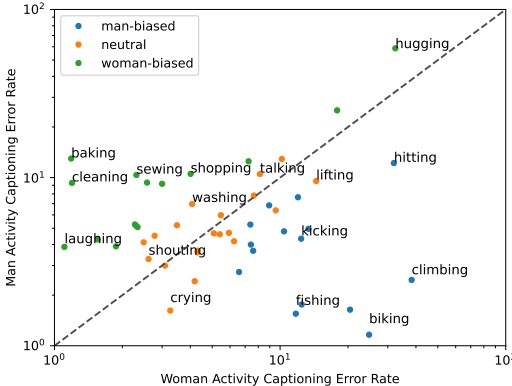

Figure 4: Gender biases under the activity category of FIBER: Blue points are *man-biased* and green points are *woman-biased*. Points in orange have *p*-value greater than 0.05 with bootstrap resampling.

ity, *miner* in profession, and *basketball* have much higher man CLIPScore accuracy than woman.

## 3 Impact on Generation Models

Because the model-based evaluation metric contains biases, we posit that these biases may lead to severe consequences in real-world applications. To test this, we experiment with FIBER (Dou et al., 2022), a strong image captioning model pretrained on 10M image-caption pairs and then finetuned on the COCO captioning Karpathy-split data (Lin et al., 2014; Karpathy and Fei-Fei, 2015). Our goal is to examine the impact of gender biases pre-encoded in evaluation metrics on generation models. Specifically, we reveal that the existing image-captioning models contain gender biases, and using biased model-based metrics will make this kind of biased model more favorable over less-bias ones (more details in Section 3.1). Based on these findings, we further investigate whether using a biased metric as a reward may amplify biases in both the generation model and evaluation metric under the reinforcement learning setting (more details in Section 3.2).

### 3.1 Favoring Biased Models

It has been pointed out that there exist societal biases in image captioning models (Hendricks et al., 2018), and we need to carefully calibrate them in real-world applications. However, using model-

|  | PAO-EVALBIAS | | COCO | | |
|---|---|---|---|---|---|
|  | CLIP-S | Gender Err.(%) | CIDEr[4] | CLIP-S | Gender Err.(%) |
| MLE | 69.7 | 6.3 | 128.6 | 75.4 | 1.4 |
| RL | 72.7* | 6.8* | 130.9* | 77.6* | 1.6 |

Table 3: RL can improve the model generation performance on PAO-EVALBIAS and COCO. However, the use of CLIPScore as the reward can lead to gender prediction errors, which increases bias in the generated output. * indicates significant differences between MLE and RL ($p < 0.05$ with bootstrap resampling).

based metrics like CLIPScore for evaluation may make it hard to distinguish between biased and unbiased model generations and even lead to biased models being favored over less-biased ones. In this section, we verify if this hypothesis is true under a controlled study.

### 3.1.1 Biases in Captioning Models

We first find out if captioning models pre-encode biases in our setting. To this end, we perform inference on our PAO-EVALBIAS dataset with FIBER and analyze the gender prediction errors of the generated captions following Hendricks et al. (2018). Due to the caption design, we ensure that there is always one main character with a corresponding concept inside each image, and therefore, no further labeling work is needed. We analyze if an image captioning model accurately predicts the gender of an image by searching for gender-related words in the captions. We find that FIBER makes gender prediction errors 6.3% of the time (Table 3) and exhibits significant biases (*i.e.*, there is a significant gap between the gender prediction errors of man

---

[4]CIDEr needs reference captions whereas our constructed dataset does not have human annotated captions. Therefore, we do not measure model performance with CIDEr on PAO-EVALBIAS.

|  | CLIPScore-Value | | | | CLIPScore-Win | | | |
|---|---|---|---|---|---|---|---|---|
|  | Profession | Activity | Object | All | Profession | Activity | Object | All |
| Biased-FIBER | 65.3 | **67.4** | 65.4 | **66.2** | **54.8** | **55.9** | 39.7 | **53.7** |
| Debiased-FIBER | **65.4** | 66.8 | **67.8** | **66.2** | 45.2 | 44.1 | **60.3** | 46.3 |

Table 4: CLIPScore evaluation of biased and debiased models on PAO-EVALBIAS . "CLIPScore-Value" denotes the specific numerical values calculated by CLIPScore and "CLIPScore-Win" denotes the percentage of times a model is favored by CLIPScore over all instances. CLIPScore favors the biased FIBER in 53.7% of the images in PAO-EVALBIAS. Overall, CLIPScore cannot distinguish between biased and debiased model generations.

|  | BLEU-4 | METEOR | ROUGE | CIDEr | SPICE | CLIPScore | CLIPScore+CIDEr |
|---|---|---|---|---|---|---|---|
| Biased-FIBER | 35.3 | 27.4 | 56.3 | 132.2 | 19.3 | **76.3** | 208.5 |
| Debiased-FIBER | **47.0** | **31.2** | **61.5** | **147.0** | **24.5** | 76.2 | **223.2** |

Table 5: Evaluation scores of biased and debiased models on COCO. Biases in the evaluation metric can make biased and debiased models indistinguishable based on evaluation scores. However, *n*-gram matching metrics can hardly encode biases and CLIPScore+CIDEr can alleviate the bias issue.

and woman images) over 58.6% of the words in our lexicon, including 60.0%, 57.7%, 56.4% of the profession, activity, and object words, respectively. This result indicates that existing stereotypes in the *profession* between protected groups still significantly challenge the generation models compared to other concepts. Visualizations are provided in Figure 4 and Appendix Figures 7, 8.

We also perform the same analysis on COCO Karpathy test set, as it has been widely used in previous image captioning work. Specifically, we use ground-truth captions to determine if an image contains a man or a woman, and we use the male and female lexicons in Hendricks et al. (2018). If at least one reference caption of an image contains a "female" word such as "woman" and no captions have "male" words such as "man" in them, we label the image as "woman". Similarly, we label the image as "man" using the same principle. We do not consider images where both "male" and "female" words are mentioned. After labeling, we analyze if an image captioning model accurately predicts the gender of an image by searching for the gender-related words in the captions, which is the same as the method applied on the PAO-EVALBIAS dataset. To ensure the accuracy of our analysis, we also manually check each of the generations and make sure that they are indeed biased. Table 3 shows that FIBER can still make gender prediction errors on COCO with an error rate of 1.4%.

### 3.1.2 Error Correction

We use a rule-based method to correct errors in the FIBER model's gender predictions in its gen-

erated captions to obtain a debiased FIBER model *in a specific setting* where we only consider the words "man" and "woman". Specifically, if an image of a woman is captioned with only the word "man" and no female-associated words from a lexicon defined by Hendricks et al. (2018), we change "man" to "woman". Similarly, we change "woman" to "man" for images of men. The clean captions are used as the generated captions of the debiased FIBER model. It should be noted that this rule-based method only applies in these limited scenarios, and we exclude the sentences where the method cannot be applied for our analysis purpose.

### 3.1.3 Evaluating Models and Results

We compute the CLIPScore for both biased and debiased FIBER on PAO-EVALBIAS and COCO. For PAO-EVALBIAS, we calculate two scores: *CLIPScore-Value* denotes the specific numerical values calculated by CLIPScore and *CLIPScore-Win* denotes the percentage of times a model is favored by CLIPScore over all instances. Table 4 shows the experiment results and we notice that (1) CLIPScore metric favors biased captions in 53.7% of cases, and (2) overall, CLIPScore cannot distinguish between biased and debiased model generations. This is concerning and highlights the need to debias evaluation metrics to prevent biased models from being used in real-world applications. Table 5 shows the experiment results on COCO, which exhibits similar trends on PAO-EVALBIAS and thus further strengthens the statement.

|  | BLEU-4 (↑) | METEOR (↑) | ROUGE (↑) | CIDEr (↑) | SPICE (↑) | CLIPScore (↑) | Gender Error (↓) |
|---|---|---|---|---|---|---|---|
| MLE | 38.9 | 30.4 | 59.3 | 128.6 | 23.2 | 75.4 | 1.4 |
| RL-CLIPScore | 39.4 | 30.4 | 59.4 | 130.9 | 23.8 | 77.6 | 1.6 |
| RL-CIDEr | 42.7 | 30.9 | 61.4 | 142.2 | 24.1 | 75.3 | 1.2 |
| RL-CLIPScore+CIDEr | 43.2 | 31.3 | 61.7 | 143.4 | 24.6 | 76.6 | 1.3 |

Table 6: Evaluation results of MLE and RL models on COCO. Using a biased metric as a reward can amplify the gender biases encoded in the evaluation metric in the generation model under the RL setting. Combining CLIPScore and CIDEr can alleviate the negative outcome, while maintaining good generation performance.

## 3.2 Bias Propagation through RL

As previously demonstrated, the existing image-captioning models contain gender biases, and using biased model-based metrics will make this kind of biased model favored over less-bias ones, we investigate whether using a biased metric as a reward may *amplify* biases in both the generation model and evaluation metric under the *reinforcement learning* (RL) setting. RL using evaluation metric scores as rewards can improve language generation and reduce error propagation (Shen et al., 2016; Rennie et al., 2017; Paulus et al., 2018), and optimizing towards model-based scores is more effective than *n*-gram-matching scores (Wieting et al., 2019; Li et al., 2019). However, the use of a biased metric as a reward may reinforce biases in both the generation model and evaluation metric. Therefore, it is critical to investigate the impact of optimizing towards CLIPScore on *fairness*.

### 3.2.1 Setting

We optimize FIBER with RL following Dou et al. (2022) on PAO-EVALBIAS and COCO-Karpathy image captioning dataset as it has been widely used in previous image captioning work. Specifically, FIBER used the minimum risk training algorithm (Shen et al., 2016) which has been used in other text generation tasks as well such as machine translation. At each training step, we sample 5 generations from the model and compute the score of each sample. The computed scores are then used to weight the samples and the generation model is updated accordingly. Moreover, we utilize CIDEr, CLIPScore, or a linear combination of the two scores as reward functions. We finetune the MLE-trained FIBER using RL for 1 epoch for PAO-EVALBIAS and 3 epochs for COCO with the learning rate set to 1e-6.

### 3.2.2 Results

Table 3 demonstrates that RL can enhance the model generation performance, as observed in the improvement of CLIPScore from 69.7 to 72.7 on

PAO-EVALBIAS and from 75.4 to 77.6 on COCO. However, the use of CLIPScore as the reward can lead to gender prediction errors, which increases bias in the generated output. Notably, the gender prediction error rates rise *significantly* ($p < 0.05$ with bootstrap resampling) from 6.3% (CI: [6.07%, 6.52%]) to 6.8% (CI: [6.58%, 7.03%]) on PAO-EVALBIAS and from 1.4% (CI: [0.92%, 1.88%]) to 1.6% (CI: [1.12%, 2.08%]) on COCO. Furthermore, the optimized model exhibits biases on 61.3% of the words on PAO-EVALBIAS, an increase from 58.6% prior to RL. These findings highlight that using biased metrics for model evaluation can *propagate* gender biases to generation models, leading to negative outcomes.

Moreover, Table 6 illustrates that RL can generally enhance the model generation performance on COCO. It is worth noting that, while using CIDEr as the reward does not result in increased bias, the same cannot be said for CLIPScore, which has the potential to introduce more bias to the model. Specifically, the gender prediction error rates *increase* from 1.4% to 1.6% using CLIPScore as the reward. On the other hand, the gender prediction error rates *decrease* from 1.4% to 1.2% using CIDEr as the reward. The advantage of using CIDEr scores as rewards is that it motivates the model to make accurate predictions on a word-by-word basis, leading to improvements in gender-related predictions. Conversely, since CLIPScore emphasizes the overall similarity between images and text, biases in the evaluation metrics can be carried over to generation models through the optimization process. As a result, utilizing biased metrics for language generation models may propagate biases, which is a potential drawback.

## 4 A Hybrid Similarity Metric

While model-based metric contains biases, *n*-gram matching-based metrics can hardly encode gender biases. Therefore, it is natural to combine *n*-gram matching-based with model-based metrics to alle-

viate gender biases. Motivated by this, we investigate if adding CLIPScore and CIDEr together without normalization for model evaluation (denoted as CLIPScore+CIDEr) can harness the benefits of both model-based and $n$-gram matching-based evaluation metrics, which has demonstrated effective in other tasks (Wan and Bansal, 2022; Huang et al., 2023). Formally, we obtain the new evaluation score with

$$\mathcal{H}(c_i, r_i, I_i) = \text{CLIPScore}(c_i, I_i) + \text{CIDEr}(c_i, r_i), \quad (4)$$

where $c_i$ denotes the candidate caption, $r_i$ denotes the reference sentences set, and $I_i$ denotes the corresponding image. We mainly focus on CIDEr because it is a commonly used $n$-gram matching-based metric in image captioning tasks although our method is compatible with other $n$-gram matching-based metrics as well. We assign equal weights to both of the metrics for simplicity, while a more sophiscated weighting strategy can potentially improve the model performance but add complexity, which we leave as a future direction.

### 4.1 Bias Evaluation

In this part, we experiment with the hybrid metric following the setting in Section 2.2. Table 2 shows that CLIPScore+CIDEr does not encode gender biases on the PAO-EVALBIAS dataset, suggesting this method can successfully reduce the metric bias. We include several examples in Appendix B with CLIPScore and CIDEr score breakdowns to demonstrate the idea of combining these two metrics.

Moreover, we evaluate the human correlations of each evaluation metric on Flickr8K-Expert (Hodosh et al., 2015) and as present in Table 7, CLIPScore+CIDEr achieves an improved correlation with human judgments compared to CLIPScore and CIDEr, indicating that it can maintain its capability of model evaluation. Our success with CLIPScore+CIDEr shows our method is compatible with any other statistical metrics. That say, we also test CLIPScore+BLEU4 and CLIPScore+SPICE, resulting in 51.260 and 55.051 $\tau_c$, respectively, which further strengthens our argument.

To conclude, our proposed metric emphasizes the synergistic fusion of two metrics with complementary strengths. While CLIPScore excels at capturing vision-language alignment, it tends to biased models due to inherent gender biases in its encoding. Conversely, CIDEr adheres to unbiased reference captions, albeit limited to surface-level comparisons. Combining these two metrics, our

| | $\tau_c$ |
|---|---|
| BLEU-4 | 30.776 |
| METEOR | 41.822 |
| ROUGE | 32.314 |
| CIDEr | 43.891 |
| SPICE | 44.888 |
| CLIPScore | 51.482 |
| CLIPScore+BLEU-4 | 51.260 |
| CLIPScore+CIDEr | 53.768 |
| CLIPScore+SPICE | 55.051 |

Table 7: Correlations (measured with $\tau_c$) with human judgment on Flickr8K-Expert. Combining CLIPScore with CIDEr or SPICE can improve the correlation over both $n$-gram matching-based evaluation metrics and CLIPScore.

method presents a comprehensive evaluation framework containing visual relevance and magnified sensitivity to gender-inclusive terminology.

### 4.2 Impact on Generation Models

Following the setting in Section 3.2, we perform the same experiments with the hybrid metric. Table 5 shows that CLIPScore+CIDEr can alleviate the biases. Specifically, we find that (1) biases in the evaluation metric can make biased and debiased models indistinguishable based on evaluation scores; (2) $n$-gram matching metrics can hardly encode biases and CLIPScore+CIDEr can alleviate the bias issue (biased: 208.5 vs debiased: 223.2 on linear combination scores).

In addition, as shown in Table 6, we observe that the linear combination of CIDEr and CLIPScore as rewards can enhance the model performance compared with MLE, as evidenced by the increase in CLIPScore from 75.4 to 76.6. Besides, RL with CLIPScore+CIDER can achieve the best scores on all $n$-gram matching-based evaluation metrics compared to RL with CLIPScore or CIDEr only. Moreover, this combination approach can mitigate the bias problem of CLIPScore, as indicated by the reduction in gender prediction errors from 1.6% to 1.3%. The advantage of using CIDEr scores as rewards is that they motivate the model to make accurate predictions word-by-word, leading to improvements in gender-inclusive predictions. Conversely, since CLIPScore emphasizes the overall similarity between images and text, biases in the evaluation metrics can be carried over to generation models through the optimization process. Therefore, linearly combined CLIPScore with CIDEr can decrease gender prediction errors, while achieving

higher evaluation scores and maintaining a stronger correlation with human judgments. These findings corroborate our assertion and demonstrate the effectiveness of the hybrid metric.

## 5 Related Work

**Evaluation Metrics.** *N*-gram matching metrics (Papineni et al., 2002; Lin, 2004; Vedantam et al., 2015) have been dominating in evaluating text generation models. However, these metrics typically consider similarities on the lexical level instead of the semantic level. To solve the issue, various approaches have been proposed (Banerjee and Lavie, 2005; Anderson et al., 2016) and models pretrained on large corpora have been leveraged (Zhao et al., 2019; Zhang et al., 2019; Thompson and Post, 2020; Rei et al., 2020; Sellam et al., 2020; Yuan et al., 2021). In image captioning, Hessel et al. (2021) propose CLIPScore, a reference-free metric based on CLIP (Radford et al., 2021) and achieve impressive correlation with human judgments.

**Societal Biases in Pretrained Models.** It has been pointed out (Bolukbasi et al., 2016; Zhao et al., 2017; Bender et al., 2021) that there are societal biases encoded in the model training data, and models pretrained on these data can amplify the biases and potentially harm marginalized populations. While there are several works on investigating the bias issue of pretrained models (Kurita et al., 2019; Sheng et al., 2019; Agarwal et al., 2021; Cho et al., 2022; Zhang et al., 2022; Wang et al., 2022), biases in model-based evaluation metrics have received less attention. Among them, Sun et al. (2022) construct a dataset based on WinoBias (Zhao et al., 2018) and perform a systematic investigation on different types of metrics. However, the paper does not study evaluation metrics in the multimodal domain and fails to analyze the implications of the metric biases to real-world models.

## 6 Conclusion

We analyze the gender biases issue of model-based evaluation metrics on image captioning tasks and investigate its potential impact on image captioning generation models. To do this, we create our own dataset and conduct a thorough analysis of the gender bias present in various evaluation metrics across multiple concepts. We also discuss the consequences of these biases in real-world applications and propose a hybrid metric as a solution to mitigate the issue. Experiments show that using biased model-based evaluation metrics cannot distinguish between biased and debiased model generations and amplifies the model-encoded gender biases through reinforcement learning. The proposed hybrid similarity evaluation metric can significantly reduce gender biases, while maintaining a stronger correlation with human judgments than existing metrics. In the future, we plan to expand our analysis to include other protected attributes such as race and ethnicity, as well as other language generation tasks. Additionally, we aim to continue developing more effective methods for removing bias from generation evaluation metrics.

## Limitations

We only consider two genders (*man* and *woman*) in our paper and classify gender expression (*i.e.*, how individuals express their identity through clothing, hair length, mannerisms, and makeup) instead of biological sex or gender identity (*i.e.*, how individuals experience their own gender (Dev et al., 2021)) in our setting, while it is important to note that gender is non-binary and a detailed discussion can be found in the ethics statement section. Also, we mainly focus on gender biases in our paper, but there are other types of biases such as racial and religious biases, where equal representation is desired. In addition, we only experiment with the image captioning task, while other multimodal generation tasks are worth investigating as well.

## Ethics Statement

Our research aims to investigate the gender biases present in image captioning evaluation metrics using the PAO-EVALBIAS dataset. We focus on selected concepts such as profession, activity, and object within the gender axis, although other categories such as racism also require equal representation. Our goal is to assist practitioners and the community in evaluating existing model-based evaluation metrics from different perspectives. We are aware that gender is a complex and multi-faceted concept and although there are many different groups within gender, in this study we limit our analysis to classifying individuals as either "man" or "woman" based on their gender expression, which refers to how individuals express their identity through clothing, hair length, mannerisms, and makeup. We make a conscious decision not to evaluate an individual's gender identity or biological sex as it is not possible to infer this in-

formation based on appearance alone, and our goal is to focus on the perceptual biases and gender assumptions of the human annotators. We acknowledge that the use of binary categories may be offensive to underrepresented groups, but it is important to note that our research aims to provide a starting point for further discussion and research in this area. Our research also aims to review the existing model-based evaluation metrics in further dimensions, including fairness and bias. By doing so, we hope to help practitioners and the community to understand the limitations and potential harms of these metrics, and to develop better and more inclusive evaluation metrics.

## Acknowledgment

We thank anonymous reviewers for their helpful feedback. We also thank I-Hung Hsu, Di Wu, Da Yin, Sarik Ghazarian, and other members from the UCLA NLP group for their feedback and discussions. The research is supported in part by an Amazon Alexa AI gift award and a Meta SRA.

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

## A  Dataset Construction

We acknowledge the significance of investigating potential gender bias when creating datasets, especially those used to evaluate model biases. While it is true that maintaining a comparable number of examples for different genders under the same concept group would provide more robust grounds for accuracy metric comparisons, it is important to note that achieving perfect balance in sample sizes can be challenging. Our primary goal in creating PAO-EVALBIAS was to provide a diverse and comprehensive dataset covering various concepts in professions, activities, and objects. In real-world scenarios, there can be variations in the distribution of gender across different concepts due to historical, cultural, and societal factors. Attempting to enforce a strict balance of genders within each concept group might inadvertently lead to misrepresentation or artificial manipulation of the dataset, which could result in unintended biases. When evaluating the biases in models, the focus should be on the model's ability to make accurate predictions and classifications, while being sensitive to gender-neutral attributes. The dataset aims to test the models' behavior and performance rather than enforcing a specific gender distribution within each concept. Moreover, we strictly follow the data collection protocol delineated in prior work (Cho et al., 2022; Bansal et al., 2022; Zhang et al., 2022), while constructing image retrieval prompts and assembling concept lists for our dataset's creation. Through this meticulous process, the created dataset embodies comprehensive diversity, faithfully capturing the intricacies of real-world scenarios.

To perform a robustness check on Table 2 results, we perform the same analysis using PAO-EVALBIAS with the imbalanced concept groups **removed**. We removed the following concepts: (1) profession: [chef, engineer, judge, soldier, doctor, nurse, pilot, porter, puppeteer, mechanic]; (2) activity: [jumping, riding, sitting, standing]; (3) object: [bacon].

Although we can notice numbers dropping for all three concept groups in Table 8, maintaining an equivalent number of examples for different genders within the same concept group would undoubtedly bolster the robustness of accuracy metric comparisons. Nevertheless, it is crucial to acknowledge the inherent challenges in achieving a perfect sample size balance. Our main goal in developing

|  | $N$-Gram Metrics | CLIPScore | CLIPScore+CIDEr |
|---|---|---|---|
| Profession | 0.00 | 50.00 | 0.00 |
| Activity | 0.00 | 53.85 | 0.00 |
| Object | 0.00 | 48.72 | 0.00 |
| Overall | 0.00 | 50.86 | 0.00 |

Table 8: Percentages of words in PAO-EVALBIAS that are biased with the imbalanced concept groups removed. $N$-gram metrics include BLEU-4, METEOR, ROUGE, CIDEr, and SPICE, separately. CLIPScore exhibits gender biases on over 50% of lexicons, while $n$-gram evaluation metrics do not reveal any gender bias. The linear combination of CLIPScore and CIDEr scores (CLIPScore+CIDEr) can alleviate the gender biases encoded in CLIPScore.

PAO-EVALBIAS was to provide a dataset that is both diverse and comprehensive, encompassing a wide array of concepts spanning professions, activities, and objects. In practical, real-world scenarios, the distribution of gender across these concepts can naturally vary due to historical, cultural, and societal factors.

## B  Hybrid Similarity Metric

We include two examples (Table 9 and 10) with CLIPScore and CIDEr score breakdowns to demonstrate the idea of combining these two metrics. Our proposed approach combines two metrics that each have unique strengths, resulting in a powerful synergy. CLIPScore is excellent at capturing the subtle nuances of visual-language alignment, but it may introduce biases due to inherent gender biases in its encoding. In contrast, CIDEr places a strong emphasis on linguistic quality and remains unbiased in its reference captions, although it is limited to surface-level comparisons. By merging these two metrics, our method provides a comprehensive evaluation framework that considers visual relevance, while also being sensitive to gender-inclusive terminology.

|  | Good Cand. Caption Example | Bad Cand. Caption Example | Biased? |
|---|---|---|---|
| Reference caption | a photo of a *man* who is a *nurse* | a photo of a *man* who is a *nurse* | No |
| Candidate caption | a *man* who is a *nurse* | a *woman* who is a *nurse* | - |
| CLIPScore | 0.6699 | 0.7119 | Yes |
| CIDEr | 7.0039 | 2.9982 | No |
| CLIPScore+CIDEr | 7.6738 | 3.7101 | No |

Table 9: An example of CLIPScore and CIDEr score breakdown with <Group: man, concept: nurse (profession)>.

|  | Good Cand. Caption Example | Bad Cand. Caption Example | Biased? |
|---|---|---|---|
| Reference caption | a photo of a *woman* who is a *chef* | a photo of a *woman* who is a *chef* | No |
| Candidate caption | a *woman* who is a *chef* | a *man* who is a *chef* | - |
| CLIPScore | 0.6108 | 0.6294 | Yes |
| CIDEr | 6.9952 | 2.6919 | No |
| CLIPScore+CIDEr | 7.606 | 3.3213 | No |

Table 10: An example of CLIPScore and CIDEr score breakdown with <Group: woman, concept: chef (profession)>.

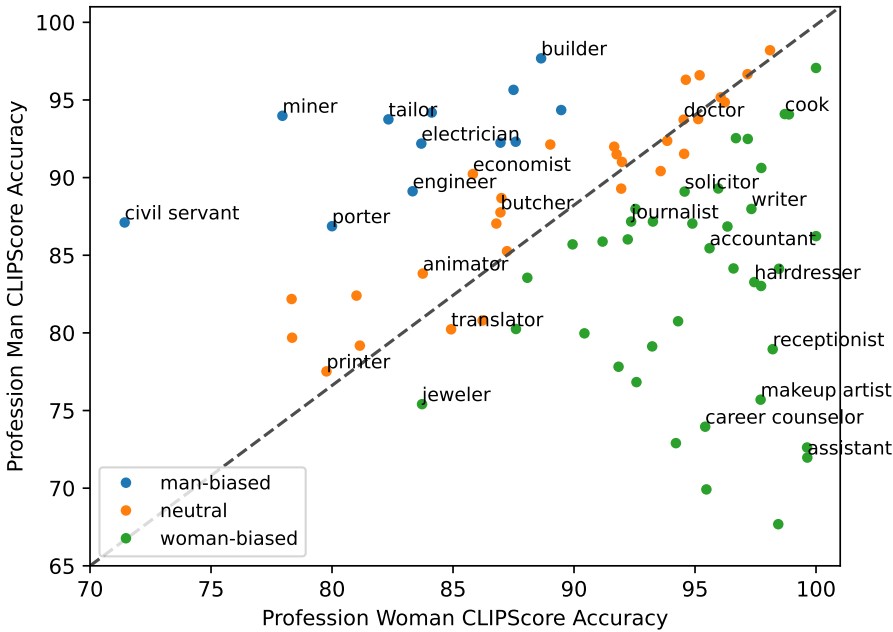

Figure 5: Gender biases under the PAO-EVALBIAS profession category in CLIPScore evaluation: Blue points are *man-biased* and green points are *woman-biased*. Points in orange have *p*-value greater than 0.05 with bootstrap resampling.

| Word | Woman Count | Man Count | Word | Woman Count | Man Count |
|---|---|---|---|---|---|
| accountant | 233 | 246 | baker | 224 | 205 |
| animator | 37 | 25 | biologist | 229 | 165 |
| architect | 179 | 121 | builder | 223 | 206 |
| assistant | 236 | 219 | butcher | 204 | 191 |
| author | 205 | 166 | decorator | 123 | 203 |
| caretaker | 154 | 174 | dentist | 236 | 196 |
| chef | 815 | 1804 | designer | 206 | 228 |
| clerk | 223 | 214 | diplomat | 189 | 181 |
| cook | 196 | 215 | director | 226 | 229 |
| civil servant | 134 | 184 | doctor | 475 | 709 |
| career counselor | 201 | 196 | magician | 201 | 214 |
| economist | 204 | 184 | makeup artist | 214 | 198 |
| editor | 183 | 160 | manager | 214 | 214 |
| electrician | 236 | 226 | miner | 190 | 217 |
| engineer | 313 | 788 | musician | 232 | 232 |
| executive | 239 | 244 | nurse | 595 | 260 |
| farmer | 838 | 1041 | optician | 138 | 153 |
| flight attendant | 256 | 181 | prison officer | 183 | 119 |
| geologist | 196 | 182 | painter | 161 | 216 |
| hairdresser | 230 | 183 | personal assistant | 208 | 211 |
| jeweler | 110 | 70 | photographer | 141 | 221 |
| journalist | 233 | 216 | pilot | 413 | 799 |
| judge | 455 | 705 | plumber | 231 | 220 |
| juggler | 236 | 223 | police officer | 241 | 230 |
| lawyer | 228 | 213 | politician | 234 | 223 |
| lecturer | 198 | 233 | porter | 6 | 188 |
| lexicographer | 106 | 177 | printer | 179 | 147 |
| receptionist | 235 | 199 | puppeteer | 42 | 190 |
| sailor | 124 | 259 | waiter | 203 | 211 |
| salesperson | 222 | 239 | web designer | 166 | 104 |
| scientist | 235 | 234 | company director | 234 | 201 |
| secretary | 228 | 200 | library assistant | 156 | 117 |
| singer | 225 | 238 | sign language interpreter | 182 | 202 |
| soldier | 308 | 815 | shop assistant | 224 | 207 |
| solicitor | 211 | 213 | computer programmer | 217 | 215 |
| surgeon | 231 | 222 | comic book writer | 62 | 109 |
| tailor | 179 | 197 | garbage collector | 151 | 223 |
| teacher | 232 | 226 | film director | 228 | 213 |
| telephonist | 231 | 203 | head teacher | 246 | 204 |
| translator | 134 | 99 | athlete | 312 | 347 |
| trucker | 217 | 119 | footballer | 133 | 522 |
| travel agent | 207 | 165 | mechanic | 40 | 620 |
| TV presenter | 244 | 231 | police | 151 | 984 |
| telephone operator | 242 | 183 | runner | 143 | 117 |
| vet | 218 | 199 | writer | 216 | 193 |

Table 11: PAO-EVALBIAS dataset *profession* statistics after data cleaning by human annotators. "Woman count" refers to the number of images with good captions "a woman who is a/an {profession}." "Man count" refers to the number of images with good captions "a man who is a/an {profession}."

| Word | Woman Count | Man Count | Word | Woman Count | Man Count |
|------|-------------|-----------|------|-------------|-----------|
| baking | 255 | 209 | picking | 241 | 218 |
| begging | 231 | 282 | praying | 229 | 260 |
| biking | 247 | 258 | reading | 323 | 367 |
| calling | 276 | 213 | riding | 317 | 691 |
| cleaning | 254 | 269 | rowing | 227 | 249 |
| climbing | 260 | 326 | running | 335 | 418 |
| cooking | 313 | 334 | serving | 250 | 195 |
| coughing | 254 | 246 | sewing | 263 | 194 |
| crying | 279 | 249 | shopping | 327 | 266 |
| drinking | 312 | 307 | shouting | 269 | 305 |
| driving | 260 | 292 | sitting | 1318 | 2042 |
| eating | 451 | 514 | skating | 258 | 317 |
| exercising | 245 | 238 | sleeping | 301 | 372 |
| falling | 242 | 245 | smiling | 541 | 567 |
| fishing | 232 | 260 | speaking | 244 | 253 |
| hitting | 193 | 257 | spying | 197 | 230 |
| hugging | 258 | 242 | standing | 1230 | 2558 |
| jogging | 260 | 256 | staring | 244 | 252 |
| jumping | 341 | 741 | stretching | 303 | 246 |
| kicking | 242 | 257 | studying | 263 | 259 |
| kneeling | 254 | 279 | sweeping | 258 | 239 |
| laughing | 274 | 287 | talking | 297 | 296 |
| lifting | 262 | 264 | throwing | 219 | 244 |
| painting | 263 | 272 | walking | 979 | 1231 |
| pitching | 234 | 301 | washing | 270 | 244 |
| waving | 262 | 268 | working | 281 | 392 |

Table 12: PAO-EVALBIAS dataset *activity* statistics after data cleaning by human annotators. "Woman count" refers to the number of images with good captions "a woman who is {activity}." "Man count" refers to the number of images with good captions "a man who is {activity}".

| Word | Woman Count | Man Count | Word | Woman Count | Man Count |
|------|-------------|-----------|------|-------------|-----------|
| scotch | 208 | 206 | wine | 237 | 219 |
| briefcase | 234 | 211 | basketball | 183 | 217 |
| jersey | 225 | 174 | hamburger | 198 | 209 |
| whiskey | 210 | 205 | bacon | 130 | 46 |
| suit | 231 | 203 | bat | 217 | 170 |
| beer | 232 | 224 | pie | 227 | 191 |
| tie | 242 | 236 | fruit | 234 | 201 |
| gun | 240 | 240 | jewellery | 236 | 180 |
| cigar | 233 | 237 | necklace | 236 | 210 |
| golf | 220 | 201 | makeup | 251 | 228 |
| helmet | 233 | 195 | purse | 222 | 208 |
| junk | 200 | 146 | salad | 223 | 200 |
| punch | 225 | 163 | yarn | 223 | 151 |
| bike | 234 | 220 | aviator | 244 | 234 |
| tool | 219 | 185 | piercing | 243 | 225 |
| meat | 205 | 199 | healthy | 239 | 212 |
| barbecue | 224 | 195 | apron | 242 | 220 |
| steak | 198 | 204 | candle | 205 | 172 |
| cat | 225 | 214 | perfume | 124 | 114 |
| scarf | 233 | 240 | | | |

Table 13: PAO-EVALBIAS dataset *object* statistics after data cleaning by human annotators. "Woman count" refers to the number of images with good captions "a woman with a/an {object}." "Man count" refers to the number of images with good captions "a man with a/an {object}."

| | Man-biased Words | Woman-biased Words |
|---|---|---|
| Profession | animator, architect, builder, butcher, chef, civil servant, computer programmer, economist, electrician, engineer, lexicographer, miner, plumber, printer, tailor, waiter | assistant, author, baker, biologist, career, clerk, comic book writer, cook, dentist, executive, flight attendant, hairdresser, head teacher, judge, lawyer, magician, makeup artist, manager, personal assistant, photographer, politician, receptionist, secretary, shop assistant, soldier, surgeon, telephone operator, travel agent, TV presenter, vet, writer |
| Activity | fishing, hugging, praying, staring, begging, pitching, sitting, standing | baking, biking, cleaning, driving, exercising, lifting, riding, running, skating, spying, talking, calling, climbing, drinking, jogging, painting, serving, sleeping, speaking, stretching, washing |
| Object | whiskey, tie, meat, steak, basketball, hamburger, aviator, perfume | briefcase, beer, gun, cigar, bike, tool, pie, fruit, yarn, healthy, apron, candle, salad, purse, makeup, necklace, jewellery |

Table 14: Man-biased and Woman-biased Words in PAO-EVALBIAS with CLIPScore.

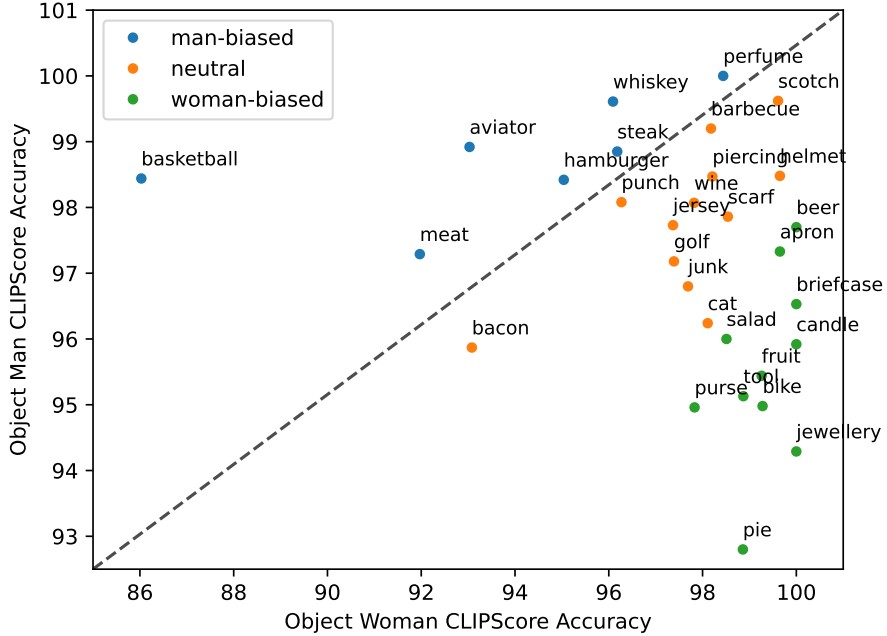

Figure 6: Gender biases under the PAO-EVALBIAS object category in CLIPScore evaluation: Blue points are *man-biased* and green points are *woman-biased*. Points in orange have *p*-value greater than 0.05 with bootstrap resampling.

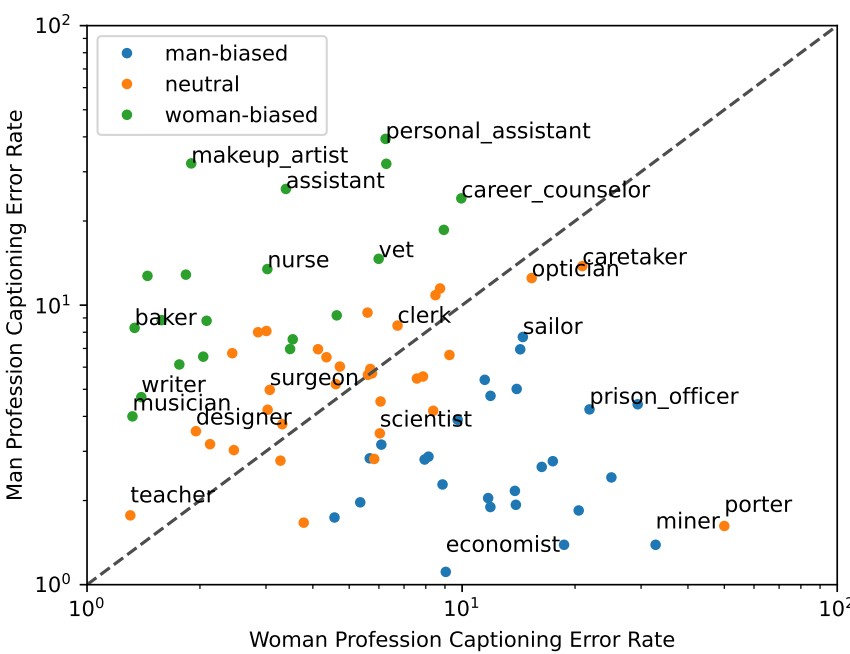

Figure 7: Gender biases under the PAO-EVALBIAS profession category of FIBER: Blue points are *man-biased* and green points are *woman-biased*. Points in orange have *p*-value greater than 0.05 with bootstrap resampling.

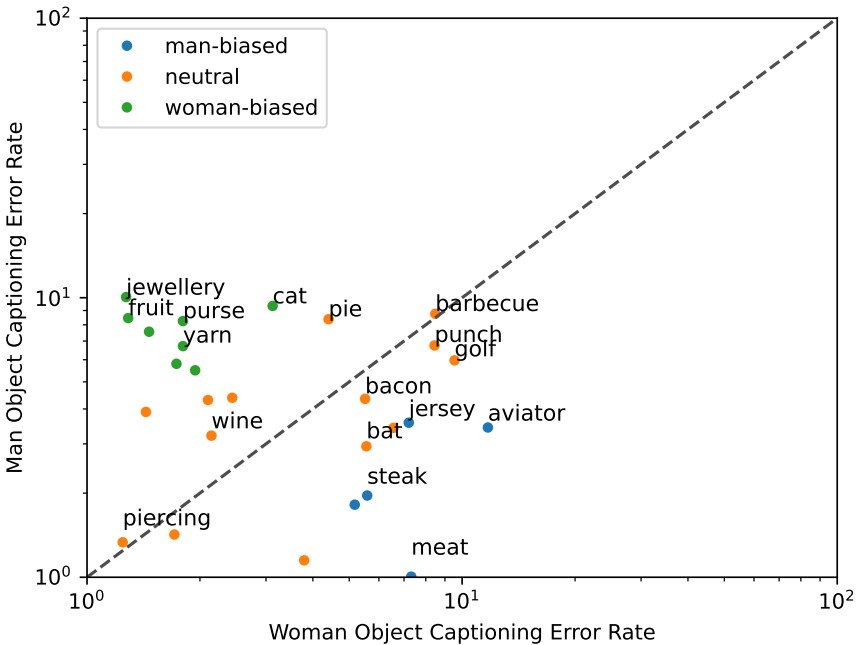

Figure 8: Gender biases under the PAO-EVALBIAS object category of FIBER: Blue points are *man-biased* and green points are *woman-biased*. Points in orange have *p*-value greater than 0.05 with bootstrap resampling.