# OpenReview forum: "Gender Biases in Automatic Evaluation Metrics for Image Captioning"
_EMNLP/2023/Conference — EMNLP 2023 Main_

### Official Review · Reviewer_FLYJ · 2023-07-21

**Soundness:** 4

**Excitement:**

4: Strong: This paper deepens the understanding of some phenomenon or lowers the barriers to an existing research direction.

**Paper Topic And Main Contributions:**

The paper proposes a new image captioning model evaluation metric such that the metric does not favor a gender-biased model across various metrics. Furthermore, the paper demonstrates that this fair metric can be used as a reward to improve the model via RL such that the resulting final model outperforms while improving the gender fairness of the model.

**Questions For The Authors:**

1. SPICE has more correlation than CIDEr with human judgement according to Table 7. Why did you use CIDEr instead of SPICE for your RL experiments recorded in Table 6?
2. When you wrote that CIDEr and CLIPScore are linearly combined, are they added with the equal weights of 1?

**Reasons To Accept:**

The paper presents a novel discovery and a novel yet easy method to improve a model's over performance and gender fairness. The method and results sound practical. The paper clearly points out the bias in the automated eval system despite its strength of better mimicking human judgement in semantic preferences. The paper is also well written and easy to follow.

**Reasons To Reject:**

n/a

**Reproducibility:**

4: Could mostly reproduce the results, but there may be some variation because of sample variance or minor variations in their interpretation of the protocol or method.

**Reviewer Confidence:**

3: Pretty sure, but there's a chance I missed something. Although I have a good feel for this area in general, I did not carefully check the paper's details, e.g., the math, experimental design, or novelty.

---

> ### Author Rebuttal · Authors · 2023-08-29
>
> We want to extend our sincere gratitude for your thoughtful feedback on our paper. Your recognition of our novel approach to enhancing model performance and gender fairness is truly appreciated. Your acknowledgment of the practicality of our method, along with your kind words about the clarity and quality of the writing, motivates us to continue pushing the boundaries of research. Thank you for your invaluable insights.
>
> ---
>
> **Questions For The Authors**
> > SPICE has more correlation than CIDEr with human judgement according to Table 7. Why did you use CIDEr instead of SPICE for your RL experiments recorded in Table 6?
>
> As highlighted in lines 447-451,  our deliberate selection of CIDEr stems from its well-established prominence as an n-gram matching-based metric in image-captioning. This preference is underscored by the fact that many recent works, such as BLIP-2 (Li et al. 2023 [1]) and mPLUG (Wang et al. 2022 [2]) have distinctly favored CIDEr as their primary evaluation metric, and CIDEr optimization is a common technique in captioning. Our success with CLIPScore+CIDEr shows our method is also compatible with other n-gram matching-based metrics.
>
> [1] Li, Junnan, Dongxu Li, Silvio Savarese and Steven C. H. Hoi. “BLIP-2: Bootstrapping Language-Image Pre-training with Frozen Image Encoders and Large Language Models.” ArXiv abs/2301.12597 (2023): n. pag.
>
> [2] Li, Chenliang et al. “mPLUG: Effective and Efficient Vision-Language Learning by Cross-modal Skip-connections.” Conference on Empirical Methods in Natural Language Processing (2022).
>
> > When you wrote that CIDEr and CLIPScore are linearly combined, are they added with the equal weights of 1?
>
> Yes, the linear combination of CIDEr and CLIPScore means they are added with the equal weights of 1.

---

### Official Review · Reviewer_KXSc · 2023-08-04

**Soundness:** 3

**Excitement:**

3: Ambivalent: It has merits (e.g., it reports state-of-the-art results, the idea is nice), but there are key weaknesses (e.g., it describes incremental work), and it can significantly benefit from another round of revision. However, I won't object to accepting it if my co-reviewers champion it.

**Paper Topic And Main Contributions:**

This paper investigates the impact of evaluation metrics on gender bias for image captioning. For this purpose, the author first constructs a dataset. In addition, the author conducted numerous experiments on different evaluation metrics, effectively demonstrating the proposed theory.

**Questions For The Authors:**

Question A: What are the advantages of your method compared to other evaluation metrics? Which needs to be emphasized in the motivation.

Question B: How do you evaluate the significance of model structure or metrics on the gender bias encoding of the model? Because you only conduct experiments in the FIBER model.

Question C:  What is the motivation behind this paper?

I'm happy to update the review if the authors can answer my concerns.

**Reasons To Accept:**

-	A newly constructed dataset and sufficient experiments.

**Reasons To Reject:**

-	This paper is a bit difficult to follow. There are some unclear statements, such as motivation.
-	In the introduction, the summarized highlights need to be adequately elaborated, and the relevant research content of this paper needs to be detailed.
-	No new evaluation metrics are proposed. Only existing evaluation metrics are linearly combined. In the experimental analysis section, there needed to be an in-depth exploration of the reasons for these experimental results.
-	A case study should be added.
-	What are the advantages of your method compared to other evaluation metrics? Which needs to be emphasized in the motivation.
-	How do you evaluate the significance of model structure or metrics on the gender bias encoding of the model? Because you only conduct experiments in the FIBER model. Furthermore, you should conduct generalization experiments on the CLIP model or other models.
-	The citation format is chaotic in the paper.
-	There are some grammar mistakes in this paper, which could be found in “Typos, Grammar, Style, and Presentation Improvements”.

**Reproducibility:**

3: Could reproduce the results with some difficulty. The settings of parameters are underspecified or subjectively determined; the training/evaluation data are not widely available.

**Reviewer Confidence:**

2: Willing to defend my evaluation, but it is fairly likely that I missed some details, didn't understand some central points, or can't be sure about the novelty of the work.

**Typos Grammar Style And Presentation Improvements:**

Typos, Grammar, Style, and Presentation Improvements:
-	In Figure 1, there is some punctuation that may be missed. (e.g., “(0.75 vs 0.72 correct)”, Should be “(0.75 vs. 0.72 correct)”, and “while the bad caption incorrectly describes the gender.” should be “, while the bad caption incorrectly describes the gender.”) In particular, the above problems appear in many parts of the paper.
-	In line 466, “That said” can be rewritten “That say”.
-	In line 493, there are missing commas. “on COCO although the difference is not significant” should be “on COCO, although the difference is not significant”. The mistake appears in many parts of the paper.
-	The structure and fluency of the sentences in the paper are poor and require improvement to enhance the quality of the writing.

---

> ### Author Rebuttal · Authors · 2023-08-29
>
> We sincerely appreciate your thoughtful review of our paper. We’re glad you found our dataset to be a valuable resource for future research and experiments. Thank you for your invaluable suggestions.
>
> ---
>
> **Reasons To Reject**
>
> > This paper is a bit difficult to follow. There are some unclear statements, such as motivation.
>
> In lines 060-112, we present detailed motivations and a straightforward workflow of our work. We notice a lack of prior work in exploring cross-modal biased evaluation metrics and understanding their implications and harm on real-world applications. Specifically, we empirically study gender biases in cross-modal metrics for image captioning. We gather a large image dataset, analyze gender biases in evaluation metrics, and their effect on models. We also present a method combining model-based and n-gram metrics to mitigate gender bias while retaining a strong correlation with human judgments for generation quality.
>
> > In the introduction, the summarized highlights need to be adequately elaborated, and the relevant research content of this paper needs to be detailed.
>
> We aim to enhance the depth of our summarized highlights and provide more detailed relevant research in the upcoming revision.
>
> > No new evaluation metrics are proposed. Only existing evaluation metrics are linearly combined. In the experimental analysis section, there needed to be an in-depth exploration of the reasons for these experimental results.
>
> Our main focuses in this paper are (1) the dataset, (2) a systematic method to analyze gender biases in automatic evaluation metrics, and how biased evaluation metrics can affect generation models.  Additionally, we propose a simple but effective method combining CLIPScore and CIDEr to reduce gender biases while maintaining high correlations with human judgments for generation quality as our other contribution. We hope our contributions can provide a good baseline and inspiration for future works. This linear combination capitalizes on the advantages of both model-based and n-gram matching-based evaluation metrics, a concept also supported by prior work [1][2] (DocNLI+ROUGE-1, FactCC+ROUGE).
>
> [1] Kung-Hsiang Huang, Hou Pong Chan, and Heng Ji. 2023. Zero-shot Faithful Factual Error Correction. In Proceedings of the 61st Annual Meeting of the Association for Computational Linguistics (Volume 1: Long Papers), pages 5660–5676, Toronto, Canada. Association for Computational Linguistics.
>
> [2] David Wan and Mohit Bansal. 2022. FactPEGASUS: Factuality-Aware Pre-training and Fine-tuning for Abstractive Summarization. In Proceedings of the 2022 Conference of the North American Chapter of the Association for Computational Linguistics: Human Language Technologies, pages 1010–1028, Seattle, United States. Association for Computational Linguistics.
>
> > A case study should be added.
>
> As a qualitative analysis, Figure 1 presents an example with an image and two candidate captions from our dataset. Notably, CLIPScore assigns a higher score to the incorrect caption (0.75 vs. 0.72 for the correct one), highlighting the presence of bias in the evaluation metric. This example underscores the potential for biased models to be favored when utilizing such biased evaluation metrics in generation tasks. We will add more examples in the revised version.
>
> > What are the advantages of your method compared to other evaluation metrics? Which needs to be emphasized in the motivation.
>
> In lines 470-486, we emphasize the driving force behind our approach by highlighting the synergistic blend of these two metrics’ complementary strengths. While CLIPScore excels in capturing visual-semantic alignment, its preference for biased models due to encoded gender biases is supported by our experiment results. In contrast, CIDEr and similar n-gram matching-based metrics prioritize linguistic quality while adhering to unbiased reference captions. Through the fusion of these metrics, our method offers a comprehensive evaluation framework that encompasses visual relevance and shines a light on gender-aware terms. To be more precise, our proposed evaluation metrics demonstrate stronger correlations with human judgment compared to other n-gram matching-based metrics (as shown in Table 7). Furthermore, employing our approach can mitigate gender biases in PAO-EvalBias and alleviate negative outcomes while maintaining high generation performance under reinforcement learning (as shown in Tables 2 and 6).
>
> > How do you evaluate the significance of model structure or metrics on the gender bias encoding of the model? Because you only conduct experiments in the FIBER model. Furthermore, you should conduct generalization experiments on the CLIP model or other models.
>
> We have experimented on BLIP [1] under FIBER's settings. Instead of testing on the whole dataset, we sample 60% of examples out of the entire dataset to conduct a quick experiment. And over 50% of the generated biased captions have higher clipscores than the debiased captions. This experimental result can support our statement on more general settings.
>
> [1] Li, Junnan, Dongxu Li, Caiming Xiong and Steven C. H. Hoi. “BLIP: Bootstrapping Language-Image Pre-training for Unified Vision-Language Understanding and Generation.” International Conference on Machine Learning (2022).
>
>
> > The citation format is chaotic in the paper.
> > There are some grammar mistakes in this paper, which could be found in “Typos, Grammar, Style, and Presentation Improvements”.
>
> We sincerely appreciate your suggestions and will make additional efforts to proofread the manuscript. We believe the copy-editing issue can be easily addressed, and we hope the reviewer can appreciate our scientific contributions.
>
> **Questions For The Authors**
>
> > Question C: What is the motivation behind this paper?
>
> In lines 035-075, we offer a comprehensive explanation of the motivations driving our work. The efficacy of pretrained model-based evaluation metrics, such as BERTScore, CLIPScore, and GPTScore, is widely recognized. These metrics have demonstrated promising performance by establishing stronger correlations with human judgments compared to n-gram matching-based metrics like BLEU, ROUGE, and CIDEr across diverse generation tasks. Unlike the conventional approach of measuring mere surface-level overlap between references and generated outputs, model-based metrics excel in capturing semantic-level similarities. Consequently, they provide more accurate assessments of model quality. Nonetheless, a prevailing consensus acknowledges that pretrained models inherently embed societal biases, including but not limited to gender, racial, and religious biases. This underscores the potential for adopting such models in evaluating generative models to amplify fairness issues inadvertently. We also notice a lack of prior work in exploring cross-modal biased evaluation metrics and understanding their implications and harm on real-world applications. As we see an increase in the variety of multimodal generation tasks such as image captioning and vision-and-language summarization [1][2], it is crucial to evaluate the cross-modal metrics specifically designed for these tasks.
>
> [1] Haotian Liu, Chunyuan Li, Qingyang Wu, and Yong Jae Lee. 2023. Visual instruction tuning. In Conference on Computer Vision and Pattern Recognition (CVPR).
>
> [2] Deyao Zhu, Jun Chen, Xiaoqian Shen, Xiang Li, and Mohamed Elhoseiny. 2023. MiniGPT-4: Enhancing vision-language understanding with advanced large language models. In Conference on Computer Vision and Pattern Recognition (CVPR).

---

### Official Review · Reviewer_g9rh · 2023-08-12

**Typos Grammar Style And Presentation Improvements:** 1. Potential inconsistency between Fi…
**Soundness:** 4

**Excitement:**

3: Ambivalent: It has merits (e.g., it reports state-of-the-art results, the idea is nice), but there are key weaknesses (e.g., it describes incremental work), and it can significantly benefit from another round of revision. However, I won't object to accepting it if my co-reviewers champion it.

**Missing References:**

I was wondering if the authors can still observe the ``gender bias" by calculating CLIPScore using FairCLIP.
[1] FairCLIP: Social Bias Elimination based on Attribute Prototype Learning and Representation Neutralization, Wang et al.

**Paper Topic And Main Contributions:**

This paper investigates the problem of gender biases in model-based automatic evaluation metrics for image captioning. In doing so, the authors construct a dataset, PAO-EVALBIAS, consisting of profession, activity, and object concepts related to stereotypical gender associations. The central idea is to first demonstrate that the model-based evaluation metric contains biases and then propose a hybrid similarity evaluation metric, a simple linear combination between CLIPScore and CIDEr, which can reduce the metric bias while not hurting its correlations with human judgments.

**Questions For The Authors:**

- Question A:
In Table 5, I was wondering about the reasons why CIDEr scores are missing for MLE and RL on PAO-EVALBIAS.

- Question B:
Could you please elaborate more on what RL algorithm you were using for the experiment in section 3.2?

**Reasons To Accept:**

- This paper presents the idea and results in a way that is relatively easy to follow and the motivation is clear and important.
- The curated dataset, PAO-EVALBIAS, consisting of profession, activity, and object concepts related to stereotypical gender associations, is a valuable resource for future research.
- The experiments are conducted in a solid way and the empirical results highlighting that the model-based evaluation metric contains biases are convincing.
- The idea of combining an n-gram matching-based evaluation metric and a model-based metric is interesting and novel. It also opens up possibilities to construct better evaluation metrics inspired by combining the best of both worlds.

**Reasons To Reject:**

- This work lacks a discussion about whether potential ``gender bias" can be introduced when the authors create the dataset, PAO-EVALBIAS. Ideally, the dataset should have a comparable number of examples for different genders under the same concept group to make the accuracy metric $Arr_{G,C}$ comparisons meaningful. However, there are still many concepts with very different sample sizes in gender groups: chef, judge, engineer, nurse, and porter to name a few.
- A more thorough analysis of the reasons why combining CLIPScore and CIDEr can mitigate gender bias is needed. It is surprising to see after the simple addition of CLIPScore and CIDEr, the percentages of words in PAO-EVALBIAS that are biased go to zero.
- The benefits of combining CLIPScore and CIDEr in the reinforcement learning setting seem to be mediocre. Is RL with CLIPScore and CIDEr significantly better than RL with CLIPScore or CIDEr only in terms of gender prediction errors?
- Although I feel that a linear combination of CLIPScore and CIDEr together is interesting, the paper doesn't fully address the motivations to do so. Why not consider a weighted average? Why can't we use CLIPScore in tandem with CIDEr in settings other than RL?

**Reproducibility:**

4: Could mostly reproduce the results, but there may be some variation because of sample variance or minor variations in their interpretation of the protocol or method.

**Reviewer Confidence:**

4: Quite sure. I tried to check the important points carefully. It's unlikely, though conceivable, that I missed something that should affect my ratings.

---

> ### Author Rebuttal · Authors · 2023-08-29
>
> We sincerely appreciate your thoughtful review of our paper. We’re glad you found our idea and results are significant. Your recognition of PAO-EVALBIAS as a valuable resource for future research is motivating. We're excited that you appreciated our strong experimental approach and compelling results. Your comments on our approach to combining evaluation metrics encourage future innovation in this direction. Thank you for your invaluable feedback.
>
> ---
>
> **Reasons To Reject**
>
> > This work lacks a discussion about whether potential ``gender bias" can be introduced when the authors create the dataset, PAO-EVALBIAS. Ideally, the dataset should have a comparable number of examples for different genders under the same concept group to make the accuracy metric Arr_G,C comparisons meaningful. However, there are still many concepts with very different sample sizes in gender groups: chef, judge, engineer, nurse, and porter to name a few.
>
> We acknowledge the significance of investigating potential gender bias when creating datasets, especially those used to evaluate model biases. While it is true that maintaining a comparable number of examples for different genders under the same concept group would provide more robust grounds for accuracy metric comparisons, it’s important to note that achieving perfect balance in sample sizes can be challenging. Our primary goal in creating PAO-EVALBIAS was to provide a diverse and comprehensive dataset covering various concepts in professions, activities, and objects. In real-world scenarios, there can be variations in the distribution of gender across different concepts due to historical, cultural, and societal factors. Attempting to enforce a strict balance of genders within each concept group might inadvertently lead to misrepresentation or artificial manipulation of the dataset, which could result in unintended biases. When evaluating the biases in models, the focus should be on the model’s ability to make accurate predictions and classifications while being sensitive to gender-neutral attributes. The dataset aims to test the models’ behavior and performance rather than enforcing a specific gender distribution within each concept. Moreover, we strictly follow the data collection protocol delineated in prior work [1][2][3] while constructing image retrieval prompts and assembling concept lists for our dataset’s creation. Through this meticulous process, the created dataset embodies comprehensive diversity, faithfully capturing the intricacies of real-world scenarios.
>
> [1] Jaemin Cho, Abhay Zala, and Mohit Bansal. 2022. Dall-eval: Probing the reasoning skills and social biases of text-to-image generative transformers. arXiv preprint.
>
> [2] Hritik Bansal, Da Yin, Masoud Monajatipoor, and Kai-Wei Chang. 2022. How well can text-to-image generative models understand ethical natural language interventions? Conference on Empirical Methods in Natural Language Processing (EMNLP).
>
> [3] Yi Zhang, Junyang Wang, and Jitao Sang. 2022. Counterfactually measuring and eliminating social bias in vision-language pre-training models. In ACM International Conference on Multimedia (ACM MM).
>
> > A more thorough analysis of the reasons why combining CLIPScore and CIDEr can mitigate gender bias is needed. It is surprising to see after the simple addition of CLIPScore and CIDEr, the percentages of words in PAO-EVALBIAS that are biased go to zero.
>
> In lines 470-486, we highlight the motivation behind our proposed approach, emphasizing the synergistic fusion of two metrics with complementary strengths. While CLIPScore excels at capturing visual-semantic alignment, it tends to biased models due to inherent gender biases in its encoding. Conversely, CIDEr prioritizes linguistic quality while adhering to unbiased reference captions, albeit limited to surface-level comparisons. Combining these two metrics, our method presents a comprehensive evaluation framework containing visual relevance and magnified sensitivity to gender-inclusive terminology. Consequently, the linear combination of these two metrics effectively mitigates gender bias, resulting in the elimination of biased terms within the PAO-EVALBIAS, as demonstrated in our settings.
>
> > The benefits of combining CLIPScore and CIDEr in the reinforcement learning setting seem to be mediocre. Is RL with CLIPScore and CIDEr significantly better than RL with CLIPScore or CIDEr only in terms of gender prediction errors?
>
> As shown in Table 6, RL with CLIPScore+CIDER can achieve the best scores on all evaluation metrics (e.g., BLEU-4: +0.5, METEOR: +0.4, ROUGE: +0.3, CIDER: +1.2, and SPICE: 0.5) except for CLIPScore. In lines 419-427, we discussed that using CIDER as the reward in RL does not result in increased bias (1.4->1.2), while CLIPScore has the potential to introduce more bias to the model (1.4->1.6). The advantage of using CIDEr scores as rewards is that they motivate the model to make accurate predictions word-by-word, leading to improvements in gender-related predictions. Conversely, since CLIPScore emphasizes the overall similarity between images and text, biases in the evaluation metrics can be carried over to generation models through the optimization process. Although RL with CLIPScore+CIDEr is not significantly better (p~0.10) than RL with CLIPScore, the small gap is also because there are not many biased generations for analysis. Therefore, linearly combined CLIPScore with CIDER can achieve the best of the worlds, decreasing gender prediction errors while achieving higher evaluation scores and maintaining a stronger correlation with human judgments.
>
> > Although I feel that a linear combination of CLIPScore and CIDEr together is interesting, the paper does’t fully address the motivations to do so. Why not consider a weighted average? Why can’t we use CLIPScore in tandem with CIDEr in settings other than RL?
>
> Our major contributions in this paper are (1) the dataset, (2) a systematic method to analyze gender biases in automatic evaluation metrics and how biased evaluation metrics can affect generation models, (3) a simple but effective method combining CLIPScore and CIDEr to reduce gender biases while maintaining high correlations with human judgments for generation quality, providing a good baseline and inspiration for future works.
>
> In lines 470-486, we underscore the motivation behind our proposed method, emphasizing the synergistic combination of these two metrics’ complementary strengths. Although CLIPScore’s advantage lies in capturing visual-semantic alignment, it prefers biased models due to encoded gender biases. Conversely, CIDEr focuses on linguistic quality while adhering to unbiased reference captions. By amalgamating these metrics, our method offers a comprehensive evaluation framework encompassing visual relevance and a spotlight on gender-aware terms. To conclude, the rationale behind this approach is to harness the benefits of both model-based and n-gram matching-based evaluation metrics, as demonstrated in prior studies [1][2] (DocNLI+ROUGE-1, FactCC+ROUGE).
>
> A weighted average could be considered, but it might not necessarily capture the intricate interactions between these two aspects as effectively as a unified metric. A linear combination allows for equal consideration of both metrics, preventing one from overshadowing the other in the final evaluation. Moreover, a weighted average might require a separate optimization step to determine the optimal weights for each metric, potentially adding complexity to the approach.
>
> Concerning the adaptability of CLIPScore+CIDEr in settings beyond reinforcement learning (RL), we recognize the versatility of this hybrid metric. We are committed to making these points more evident in the revised version.
>
> [1] Kung-Hsiang Huang, Hou Pong Chan, and Heng Ji. 2023. Zero-shot Faithful Factual Error Correction. In Proceedings of the 61st Annual Meeting of the Association for Computational Linguistics (Volume 1: Long Papers), pages 5660–5676, Toronto, Canada. Association for Computational Linguistics.
>
> [2] David Wan and Mohit Bansal. 2022. FactPEGASUS: Factuality-Aware Pre-training and Fine-tuning for Abstractive Summarization. In Proceedings of the 2022 Conference of the North American Chapter of the Association for Computational Linguistics: Human Language Technologies, pages 1010–1028, Seattle, United States. Association for Computational Linguistics.
>
> **Questions For The Authors**
>
> > Question A: In Table 5, I was wondering about the reasons why CIDEr scores are missing for MLE and RL on PAO-EVALBIAS.
>
> CIDEr needs reference captions whereas our constructed dataset does not have human annotated captions. Therefore, the CIDEr scores calculation are not suitable for MLE and RL on PAO-EVALBIAS.
>
> > Question B: Could you please elaborate more on what RL algorithm you were using for the experiment in section 3.2?
>
> We followed FIBER and used their public codebase for RL training (line 391-392). Specifically, FIBER used the minimum risk training algorithm (Shen et al, 2016) which is widely used in text generation tasks such as machine translation. The general idea is that at each training step, we sample 5 generations from the model and compute the score of each sample. The computed scores are then used to weight the samples and the generation model is then updated accordingly.
>
> **Missing References**
>
> > I was wondering if the authors can still observe the ``gender bias" by calculating CLIPScore using FairCLIP. [1] FairCLIP: Social Bias Elimination based on Attribute Prototype Learning and Representation Neutralization, Wang et al.
>
> We appreciate your suggestion to explore the FairCLIP paper. After thoroughly reading it, we see that FairCLIP can be used as a general debiasing method for other fairness issues related to CLIP. We plan to modify CLIPScore based on FairCLIP and conduct more experiments to investigate gender biases. Unfortunately, the unavailability of the FairCLIP code is a challenge to our efforts to replicate and verify outcomes within our limited timeframe. Nonetheless, we will cite the FairCLIP paper as a reference and are enthusiastic about future experiments to enrich our understanding of this pivotal topic.
>
> **Typos Grammar Style And Presentation Improvements**
>
> Thank you for your suggestions on improving our presentations for better understanding. While we acknowledge the possibility of bias in model-based evaluation metrics stemming from pre-trained models, we cannot confirm that CLIPScore inherits biases from model training data. Other factors, such as model architecture, may also contribute to biased evaluation metrics. This discussion is beyond our scope since we focus more on the potential bad consequences of using biased model-based evaluation metrics. We are committed to making these points more evident in the forthcoming revision.

---

### Meta-Review · Area_Chair_f7Fo · 2023-09-20

**Recommendation:** 5

**Metareview:**

This paper outlines a systematic analsys aimed at studying automatic gender bias evaluation metrics in the context of image captioning tasks. The authors released a dataset to measure this issue and also introduce a novel evaluation to reduce metric bias while mantaining correlations with human judjements.

I agree with the positive comments of Reviewer 3 and I believe the authors strongly replied to the critiques of the reviewers, especially Reviewer 1. I strongly encourage the authors to refine the paper, taking into account the feedback provided by all the reviewers, especially Reviewer 1.

---

### Decision · Program_Chairs · 2023-10-07

**Decision:**

Accept-Main

**Comment:**

This paper outlines a systematic analsys aimed at studying automatic gender bias evaluation metrics in the context of image captioning tasks. The authors released a dataset to measure this issue and also introduce a novel evaluation to reduce metric bias while mantaining correlations with human judjements.

I agree with the positive comments of Reviewer 3 and I believe the authors strongly replied to the critiques of the reviewers, especially Reviewer 1. I strongly encourage the authors to refine the paper, taking into account the feedback provided by all the reviewers, especially Reviewer 1.